# A Smart Glucose Monitoring System for Diabetic Patient

**Amine Rghioui [1,*], Jaime Lloret [2], Mohamed Harane [1] and Abdelmajid Oumnad [1]**

[1] Research Team in Smart Communication-ERSC-Research Centre E3S, EMI. Mohamed V University in Rabat, Rabat 10000, Morocco; med_mhfd.harane@research.emi.ac.ma (M.H.); aoumnad@emi.ac.ma (A.O.)

[2] Integrated Management Coastal Research Institute, Universitat Politecnica de Valencia, 46370 Valencia, Spain; jlloret@dcom.upv.es

\* Correspondence: rghioui.amine@gmail.com

**Abstract:** Diabetic patients need ongoing surveillance, but this involves high costs for the government and family. The combined use of information and communication technologies (ICTs), artificial intelligence and smart devices can reduce these costs, helping the diabetic patient. This paper presents an intelligent architecture for the surveillance of diabetic disease that will allow physicians to remotely monitor the health of their patients through sensors integrated into smartphones and smart portable devices. The proposed architecture includes an intelligent algorithm developed to intelligently detect whether a parameter has exceeded a threshold, which may or may not involve urgency. To verify the proper functioning of this system, we developed a small portable device capable of measuring the level of glucose in the blood for diabetics and body temperature. We designed a secure mechanism to establish a wireless connection with the smartphone.

**Keywords:** healthcare; data classification; machine learning; diabetic patient monitoring

## 1. Introduction

Technologies and application of the Internet of Things are advancing considerably in important IoT technical areas, becoming more accessible, available and versatile every day, allowing faster growth of objects interconnected via the Internet [1]. The quality of life of patients is an important objective of intelligent healthcare and is a widespread concern with the construction of new applications in this field. The daily mobile health service is becoming more and more important. Several chronic diseases, like cardiovascular and diabetic diseases, influence the health of living people [2,3].

Patient data is very useful. Indeed, data analytics can be applied to identify people who need "proactive care" or who need a lifestyle change in order to avoid deterioration in health. For example, patients at an early stage of certain diseases (like heart failure, which is often caused by certain risk factors such as hypertension or diabetes) should be able to benefit from predictive and preventive care thanks to big data [4]. Patients are also more comprehensive about giving away part of their privacy if this could save their lives or other people's lives [5,6].

Diabetes is a global disease which is caused by an increase or decrease in the glucose level in blood. If this abnormality persists for a long period of time, the diabetic patient has a high chance of being affected by other health problems, like damage to the blood vessels, or the risk of heart problems.

The World Health Organization (WHO) estimates diabetes to be the seventh-most-lethal disease [7,8]. Diabetes is a very dangerous disease that affects the majority of people, and a lot of attention is needed to keep them healthy. This disease is becoming one of the leading causes of death worldwide. A statistical report revealed that around 61.3 million people are recognized as diabetic patients, and the count may increase to 102 million by 2030 approximately [9]. Unfortunately, most people do not know

the importance of keeping their diabetic health stable. The main cause of diabetes is the increase or decrease in the level of glucose in the blood, and it is necessary to keep it stable in a specific interval. Some patients with abnormalities require intensive care and monitoring to avoid worsening [10,11]. The increase in the number of diabetic patients also increases the use of continuous glucose monitoring devices, which are becoming a new method of monitoring glucose levels. These devices provide real-time information on glucose levels that are updated every ten minutes.

In this paper, we develop a wireless glucose monitoring system on a smartphone using a Wi-Fi protocol for communication. The blood glucose readings taken each time will be sent regularly to the mobile phone via Wi-Fi and will be shared on the Android application. The objective aims at discussing the realization of a digital blood glucose meter system integrated into Wi-Fi in order to monitor the blood sugar level of a diabetic patient in real time. Each time the patient makes glucose level measurements, the device will automatically send the measured data to medical experts. This technology is expected to help patients; this technology should help patients to communicate periodically with their doctors. Therefore, the doctor can always monitor the health of their patients remotely to provide preventive treatment and to intervene in an emergency.

Smart devices such as smartphones and smart gadgets have several internal sensors that can help monitor vital signs in humans, such as heart rate, respiratory rate, and respiratory changes, among others [12]. They also contain a GPS which can help monitor the physical activities and the position of the patients [13].

In this article, we show a new smart system for monitoring diabetic patients. Using the data collected by the glucose sensor, diabetic patients can be monitored remotely by doctors. We have developed an intelligent algorithm, which is able to send the glucose data measured by the sensor to the doctors to analyze, these data will be automatically stored in the database as positive or false positive after being validated (or not) by the doctor. In our case study, we used several sensors to facilitate monitoring of the diabetic disease.

The remainder of this paper is organized in the following manner. Section 2 presents related works. Section 3 describes our proposed scenario for monitoring diabetic patients, as well as important design considerations. Section 4 details the hardware design for our system and the implementation of our system. Section 5 describes the experimental setup, as well as the results of our practical evaluations. Finally, Section 6 concludes the paper.

## 2. Related Work

This section includes some existing research related to some applications of glucose monitoring in diabetic patients based on the use of intelligent sensors.

We will now detail some of the work that uses sensors integrated into the smartphone for patient monitoring [14–16]. A new intelligent collaborative system that uses smartphone sensors to monitor disabled and elderly people has been proposed by Sendra et al. [17]. Wang and Lee [18] are developing a new concept of a blood glucose sensor for advanced control and monitoring of blood sugar in patients.

Ketabdar and Lyra [19] proposed a new system for monitoring and tracking the physical activities of the elderly or disabled. The system uses smartphones to collect data. The researchers performed a test with 320 activity instances and four subject users were recorded and analyzed. The presented system had an accuracy of 92.9%.

L. Shi et al. [20] suggested a new system for elderly and chronically ill patients. The system allows to the patients to control and supervise their situation. With the help of the proposed system, which is implemented in a smartphone, the patient can stay at home independently to avoid hospitalization for a long time. The system can detect a failed measurement. The communication of the data to the server is ensured by the TCP/IP protocol and the Wi-Fi technology.

Ahmed [21] presents a system for predicting the concentration of glucose for a diabetic patient. They use GlucoSim software to analyze information from the diabetic patient, which is generated

by a continuous glucose monitoring system (CGS). The CGM sensor uses a Kalman filter (KF) to reduce noise.

In [22], S. A. Siddiqui and their colleagues present a survey of non-invasive/pain-free blood sugar control methods. They also make a comparison between the different devices manufactured for non-invasive monitoring.

Lee and Chung [23] present a wearable smart shirt for ubiquitous health and activity monitoring using the wearable sensors with a smartphone. Their proposed system aims to provide continuous and real-time monitoring for patients, using the electrocardiogram and acceleration signals in a wireless body area sensor network (WBAN).

González-Valenzuela et al. [24] presented a new system for the medical monitoring of patients at home. This system is capable of obtaining data on the health of patients. The system consists of four sensors placed in the patient's body. The authors have shown the use of sensors in the body under different activity models.

We will use the data analysis system for diabetic patients, which can help solve one of the problems of the cost of patients for individual treatment. Several researchers have developed methods of data analysis and classification [25,26]. In this work, we use the random tree, naïve Bayes, SMO, ZeroR, OneR, and J48 classification algorithms from the diabetes dataset to identify the most powerful algorithm for determining the risk level for diabetic patients.

There are many proposals for elderly people and for disabled people in the mentioned works. This work suggests the implementation of a blood glucose measurement system with a Wi-Fi module for telemonitoring, but no one of the mentioned research proposals is specially designed and developed for diabetic patient monitoring. Several applications use Bluetooth, but our application uses Wi-Fi for communication. There are many proposals for elderly people and for disabled people. We do not find any applications that have already been used or are ready to use in the market for the glucose monitoring systems. The results shown in Table 1 indicate that our application is one of the applications that use Wi-Fi as a means of communication and that it is equipped with many types of sensors to collect data related to movements, body temperature, and glucose. In addition to this, our application classifies this data. In this paper, we propose our glucose level monitoring application for diabetic patients, but we are not limited to monitoring; we also apply the algorithms for classification of the data recovered by our application to classify and analyze them.

**Table 1.** Comparisons of the proposed application and other state-of-the-art applications.

| Authors | Application | Use Smartphone | Sensor's Communication | Data Classification |
|---|---|---|---|---|
| Rghioui, Lloret | diabetic patient monitoring system | Yes | Wi-Fi (Esp8266) | Yes |
| Ghasemi [15] | elderly person's health-care system | Yes | Bluetooth/Zigbee | No |
| Burton [16] | Smartphone for monitoring basic vital signs | Yes | NFC technology | No |
| Ketabdar [19] | Using Mobile Phones in Live Remote Monitoring of Physical Activities | Yes | Wi-Fi/GPRS | No |
| Siddiqui [22] | Blood Glucose monitoring | Yes | not specified | No |
| González-Valenzuela [24] | healthcare monitoring system of convalescing patients at home | No | not specified | No |
| Gomez [27] | Patient Monitoring System Based on Internet of Things | Yes | not specified | No |

## 3. Proposed Scenario

In this part, we give a detailed description of the proposed monitoring system for diabetic patients.

*Architecture*

In this section, we will develop a health care surveillance system to help diabetic patients react to their situation and manage their illness themselves. The proposed system automatically records several data related to the health of diabetic patients. In addition, the system will be seen as a platform between patients and doctors for information on daily health care.

Our proposed system consists of three main components: (i) sensor modules, (ii) a data acquisition module and (iii) a database server, which is a mobile application that works like a local server installed in the smartphone. Figure 1 shows the proposed architecture for the diabetic monitoring system.

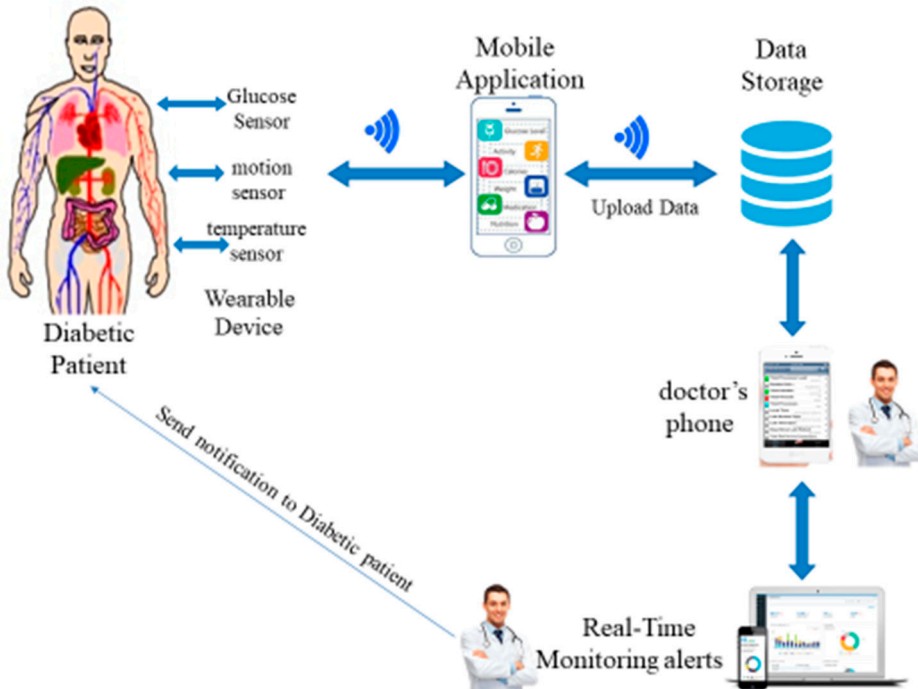

**Figure 1.** Proposed architecture for diabetic monitoring.

The architecture of our proposed system consists of three components: (i) sensor modules, (ii) a data acquisition module and (iii) a database server:

(i)    Wearable Device. This is composed of several sensor modules, like a glucose sensor, a motion sensor and a temperature sensor, and is controlled by Arduino Uno and connected to a smartphone. We used Bluetooth to connect sensor devices with the smartphones. We can use also smartwatches or gadgets that usually communicate with a smartphone using Bluetooth. The body sensors collect information about glucose level, temperature and motion from the diabetic patient. This information is sent to the mobile device through a Bluetooth connection. The mobile device sends this information to the database through the 4G network or Wi-Fi.

(ii)   Data acquisition module. This is composed of the mobile application and database server. The database server collects the data from the sensors; we used the 4G network to send the sensors' information to the database server.

(iii)  The processing unit. This is composed of a doctor's mobile and monitoring systems. The smartphone sends information to a processing unit via a 4G network. The monitoring system analyses the data gathered from the sensors. When the system detects an abnormal

situation, a notification is sent to the doctor to see the mobile application and detect if there is an emergency or just a perturbation of the use of medication with the diabetic patient.

Each diabetic patient can carry a portable device or a developed device attached to the hand, which is connected with the telephone of the patient. These devices send signals with different types of information—temperature, movement, location, and blood glucose level—to update the health profile of the diabetic patient in real time.

The doctor, the patient, and their family members can access the monitoring system to monitor physical activity and monitor the glucose level of diabetic patients using their connected smartphone. We will create a special account with a personal access code for family members of the diabetic patient in order to prevent unauthorized access to the data.

## 4. Hardware Design and Implementation

The overall structure of hardware used in our proposed architecture for monitoring the diabetic patient is illustrated in Figure 2.

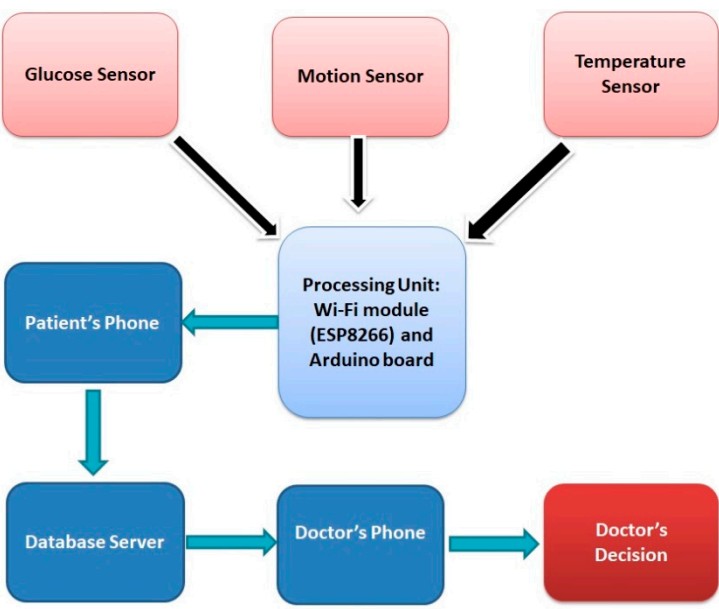

**Figure 2.** Hardware Block Diagram.

### 4.1. ESP 8266 Mod

The ESP8266 is a microcontroller integrated circuit with a Wi-Fi connection. The modules incorporating this circuit are widely used to control devices over the Internet. ESP8266 can be programmed in several ways: in C++, with the Arduino IDE3; in JavaScript, with the firmware Espruino4; and in C, with the SDK of Espressif. ESP8266 has become a popular and affordable solution for Internet of Things applications.

The ESP8266 mod uses Wi-Fi technology, which offers advantages like precise location services to within approximately a centimeter. Sending data across obstacles and walls must also be taken into account.

### 4.2. Glucose Sensor

Glucose sensors are used to measure the blood glucose concentration of a patient; this is part of a continuous glucose monitoring (CGM) system that is inserted under the skin and measures the glucose levels. It sends the glucose data wirelessly to the system receiver or a compatible smart device.

### 4.3. Pedometer

The pedometer is used to monitor the physical activity of the diabetic patient. Physical activity is defined as any movement that increases the use of energy, such as walking or playing a football game [28]. Physical activity should be recommended to all diabetic patients as part of the management of their level of glucose and for improving their health. For our practical application, we tried to work with the smartwatch, which already contains a pedometer. We ask the diabetic patient to put the smartwatch in their hands, and we automatically receive the number of steps and the distance crossed by the patient on the application installed on the smartphone of the patient.

Figure 3 shows the smartwatch and the mobile application used to record the number of steps traveled by the diabetic patient.

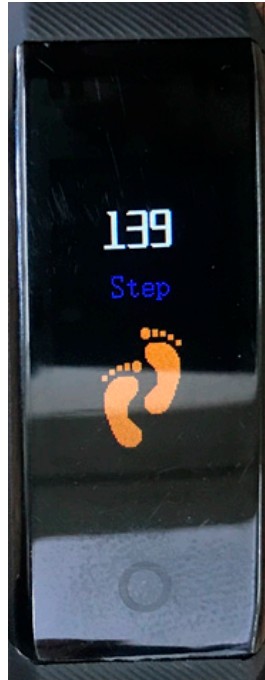 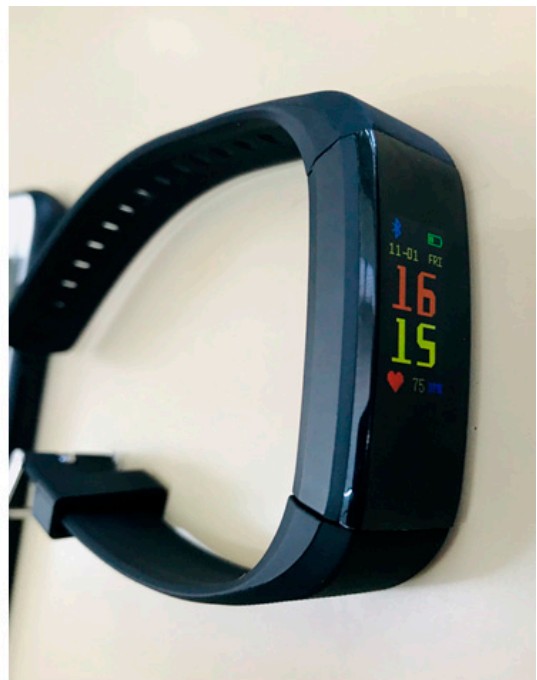

**Figure 3.** Pedometer.

### 4.4. Arduino Nano

The Arduino board is a reliable element capable of running a complete program. It represents a suitable solution for controlling electronics. The Arduino Nano board is a small electronic board capable of running a full program to perform a variety of tasks. It is a platform based on a simple input/output interface. In addition, the Arduino board has serial communication interfaces; it works with a mini-USB cable instead of a standard cable, which is also used to load programs from computers.

The data collected by the sensors is sent and processed by the microcontroller of the Arduino board.

The level of glucose in the blood (in mg/dL) of each patient was checked and is shown in Table 2, with an average of three measures daily for 10 days, and the average of number of steps per day.

**Table 2.** Glucose, temperature and number of step measurement.

| Day | Blood Sugar Level (mg/dL) | | | Temperature | Number of Steps |
|---|---|---|---|---|---|
| | **Morning** | **Afternoon** | **Evening** | | |
| Day1 | 108 | 121 | 131 | 37 | 3423 |
| Day2 | 166 | 123 | 124 | 37 | 4322 |
| Day3 | 103 | 112 | 114 | 37 | 4876 |
| Day4 | 134 | 102 | 98 | 37 | 4657 |
| Day5 | 141 | 72 | 88 | 38 | 4511 |
| Day6 | 150 | 147 | 123 | 36 | 4690 |
| Day7 | 69 | 78 | 82 | 38 | 8768 |
| Day8 | 111 | 100 | 111 | 37 | 4121 |
| Day9 | 98 | 87 | 86 | 37 | 7823 |
| Day10 | 76 | 70 | 77 | 38 | 8543 |

This part presents the implementation of our proposed system with the scheme and the flowchart of the system shown in Figure 4.

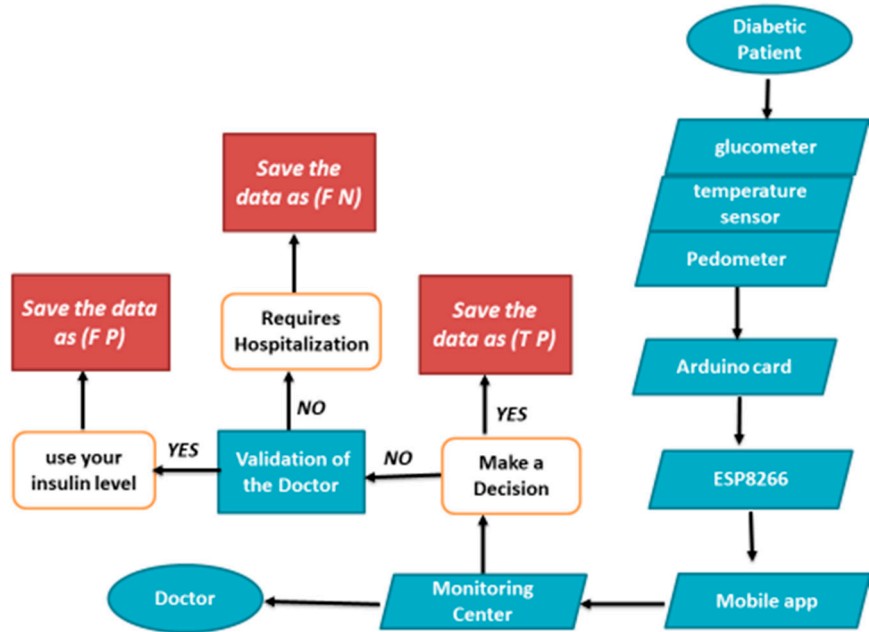

**Figure 4.** Flowchart of the system.

Data are collected for the smart diabetic monitoring system (glucose sensors, temperature sensor, pedometer), and the information is then transmitted through the ESP8266 module via a Wi-Fi network to the mobile application. These data, once pre-processed, are sent to the monitoring center for rule-based processing to produce relevant results. Having obtained the processed results, the monitoring center communicates with the doctor (via the doctor's phone), sends their decisions via a Wi-Fi network, and reports the diabetic patient state. After that, the system saves the data automatically. Finally, we can save the data as:

- False Negative (FN), for the critical data and requires hospitalization for the patient.
- True Positive (TP), for the correct data.
- False Positive (FP), for the data that requires just the injection of the necessary insulin level.

To automatically detect emergency situations in our proposed system, we have described the decision algorithm, which combines the sensor data used in real time. Figure 5 shows the intelligent algorithm for monitoring diabetic patients. This intelligent algorithm is created to decide the actions to

be taken after the data storage and analysis. This intelligent algorithm is able to detect if the data are FN, TP or FP. If the data save as FN it indicates the beginning of an emergency situation, and then the doctor verification process will start as soon as possible, as is shown in Figure 5.

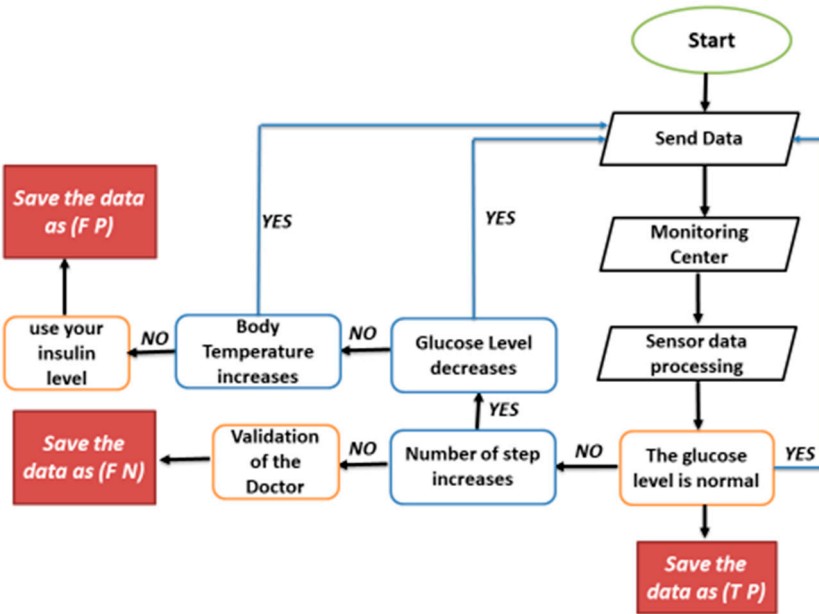

**Figure 5.** Smart algorithm for monitoring diabetic patients.

## 5. Communication Protocol

Our proposal is based on diabetic patient monitoring. We consider mobile nodes (installed in the patient's body) which have the role of taking data from the patient continuously. In particular, the installation of wireless nodes is based on module Wi-Fi (ESP8266). With this, we can monitor the evolution of parameters such as glucose level, temperature, and physical activities of diabetic patients and combine all these data to build a real-time application in healthcare to monitor the diabetic patient. Our proposed system registers the data of the patient: their name, glucose level, temperature, and other information on their physical activity. This data is then forwarded to the database. If an abnormal situation is detected, the unfavorable results are displayed, and a log is created in the database with the hour of the detection.

The wearable device is based on an Arduino Nano Board and ESP8266 Wi-Fi interface. The ESP8266 Wi-Fi module is used to connect with the mobile phone. For the part of sensors, we have selected a glucometer to measure the glucose level for diabetic patients, a temperature sensor to measure the corporal temperature and a pedometer that is able to measure the physical activities for each patient. The proposed system receives information from several sensors. This information is transmitted via Wi-Fi to a smartphone that will be in charge of sending this information to the doctor and the database at the same time. The data generated by the sensors will be sent to the database for treatment and sent to the doctor to make a decision and will be sent to the patient's family for more information on the patient's state of health. The transmission will be either from sensor to mobile phone, from phone to the database, and from the database to the doctor.

√ The transmission between the sensors and the smartphone: we have captured information about the patient (glucose level, temperature, number of steps) from each sensor; for each sensor, we must save the ID, the patient name and the transmission time. We need also to know the ID of each sensor and identify it because the same patient can wear several sensors of the same type.

√ The transmission from the smartphone to the database: the purpose of this transmission is to transform the data from the telephone to the database in order to process and classify it with the Weka tools. The transmission will be made by the Wi-Fi protocol or the 4G connection.

√ The transmission from the database to the doctor: This module is divided into two parts: the glucose level classifier, which indicates whether the glucose level detected by the glucometer is critical or not; and the doctor, who decides the type of action to be performed by the patient in order to guarantee good medical conditions for the diabetic. Figure 6 shows the structure of information transmission.

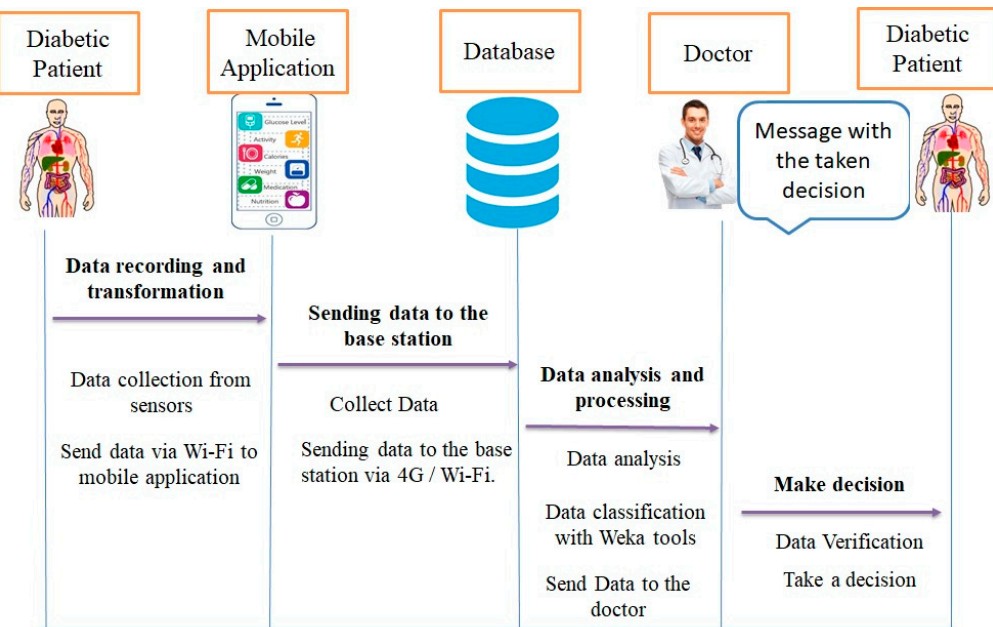

**Figure 6.** Structure of information transmission.

Figure 7 illustrates the flowchart of the proposed application, to classify the blood glucose level data for diabetic patients. More concretely, this lets the reader know how the actual glucose values are measured and sent to the base station and how to automatically process them to send them to the doctor to give the final decision. The database compares, at all times, the level of glucose received with respect to the given threshold. If a measurement is above or below the threshold, we get an information message. The next step is to contact the doctor to find out if the message contains a high risk that requires the doctor's intervention or not.

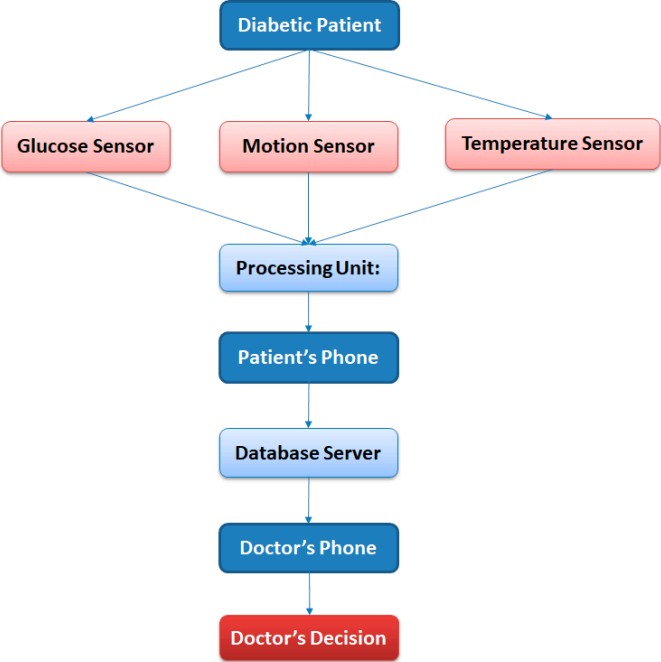

**Figure 7.** Flow Chart of Programming.

To analyze the network flow of our application, we will use the Wireshark application [29]. Wireshark is a free and open-source package analyzer. It is used in troubleshooting and analysis of computer networks, protocol development, education, and reverse engineering. Figure 8 shows us the packet captured during the transmission between ESP8266 and Arduino.

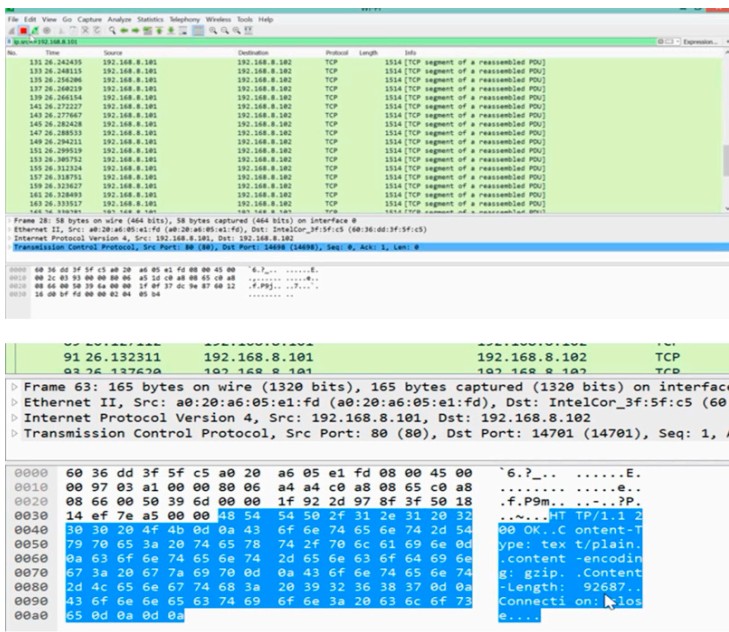

**Figure 8.** The packet captured during transmission between ESP8266 and Arduino.

Wireshark IO Graphs will show you the overall traffic seen in a capture file, which is usually measured in rate per second in bytes or packets. The ESP8266 microcontroller can be programmed in Arduino. An easy-to-use library for generating Wireshark-readable PCAP files can be loaded onto an ESP8266. The x-axis is the tick interval per second, and the y-axis is the packets per second. Figure 9 shows the IO graphs for the capture file.

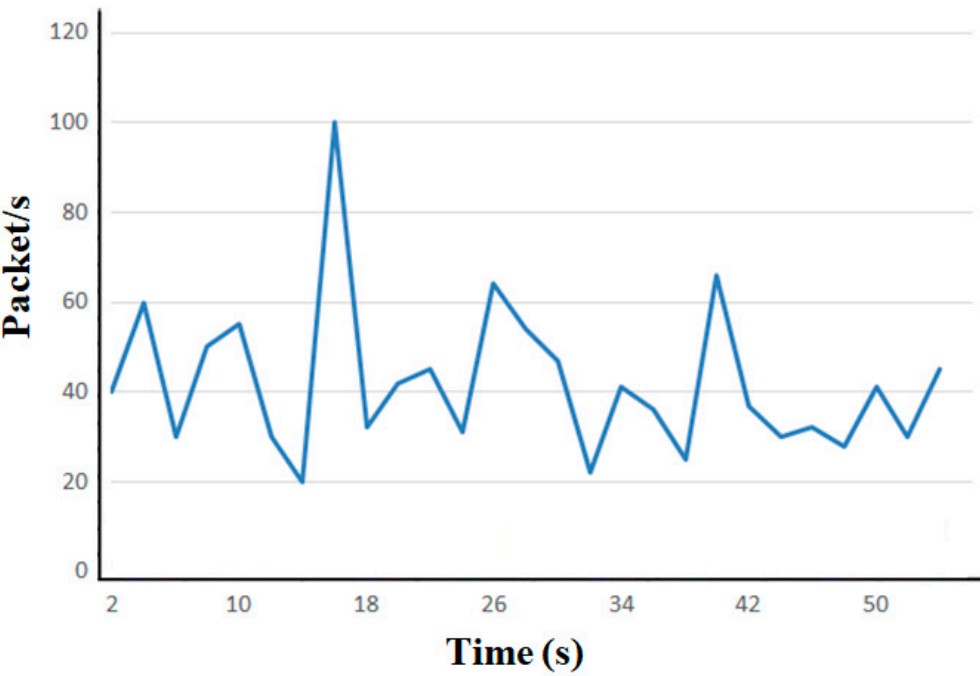

**Figure 9.** The I/O Graphs.

## 6. Data Classification

This article is designed to study the level of glucose in the blood for diabetic patients. Data were collected from three patients in the study of this case. Table 3 shows the glucose monitoring for a diabetic patient three time per day.

**Table 3.** Glucose level storage for a patient.

| Name | Device Id | Seq-Num | Glucose Level (mg/dl) | Date | Time |
|------|-----------|---------|-----------------------|------|------|
| Badr | F024275G0723 | 1 | 112 | 13/07/2019 | 10:14:53 |
| Badr | F024275G0723 | 30 | 118 | 22/07/2019 | 10:15:43 |
| Badr | F024275G0723 | 57 | 92 | 32/07/2019 | 10:07:02 |
| Badr | F024275G0723 | 71 | 105 | 05/08/2019 | 22:30:37 |
| Badr | F024275G0723 | 128 | 82 | 28/08/2019 | 15:18:16 |
| Badr | F024275G0723 | 165 | 138 | 01/09/2019 | 14:22:44 |
| Badr | F024275G0723 | 251 | 108 | 28/09/2019 | 22:46:06 |
| Badr | F024275G0723 | 312 | 72 | 14/10/2019 | 15:43:07 |

With actual new technologies, it is possible to implement an IoT solution for monitoring diabetic patients. The ESP8266 module can be considered the best choice for sending the continuous data of each sensor. In our proposal, we use portable sensors, such as the glucometer, temperature sensor, and pedometer to monitor the vital signs data of diabetic patients. The communication between the sensors and a smartphone was made using a Wi-Fi protocol.

In this paper, we developed a new glucose monitoring system for diabetes using Wi-Fi. This system can generate glucose data for diabetic patients and send this data to the data center for classification. We applied several data classification algorithms to classify glucose data; after several tests, the evaluation result showed that the system using the J48 algorithm exhibited excellent classification with the highest accuracy of 99.17%, a sensitivity of 99.47% and a precision of 99.32%. Therefore, the J48 is one of the state-of-the-art techniques to address classification problems and it provides better accuracy. The method proposed in this paper aims at providing an optimal solution for the classification of diabetic patients according to their temperature and physical activity.

The main idea of this study (diabetes classification) is to allow an early prediction of abnormal diabetes cases that require a doctor's intervention, using data on blood glucose level, temperature, and the number of steps traveled by patients, taken by sensors connected with a smartphone based on Wi-Fi technology. An early prediction of the disease can lead to treating patients before the disease gets worse and becomes critical. All of the data studied include the records of 55 diabetic patients (39 men and 16 women), for 65 days with an average of three measurements per day.

The classification algorithms used in the study were the naive Bayes (NB), J48, random tree, ZeroR, SMO (sequential minimal optimization), and OneR algorithms. In this experiment, we used WEKA (Waikato Environment for Knowledge Analysis: is an open-source data mining program), as a data extraction tool to classify data and analyze and determine the accuracy of prediction of different data mining algorithms [30].

In this study, the glucose level dataset was classified by several classification algorithms in the same data-mining programs. The glucose level dataset contains seven attributes and 10,725 glucose level records. Table 4 shows the data type and description of the attributes.

**Table 4.** Glucose Data.

| No. | Attributes | Data Type |
|-----|------------|-----------|
| 1 | Sex | Boolean |
| 2 | Age | Numeric |
| 3 | Day | Numeric |
| 4 | Glucose Level | Numeric |
| 5 | Number of Step | Numeric |
| 6 | Temperature | Numeric |
| 7 | BGL | Boolean |

## 7. Result and Discussion

Performance results in terms of precision, receiver-operating characteristics (ROC) and accuracy were analyzed and discussed.

The accuracy of the classification is defined as an evaluation measure to compare the results obtained by several methods applied in the dataset. The accuracy of the classification can be given by the equation [31].

Accuracy:

$$\frac{\sum_{i=1}^{|N|} Evaluate\ (i)}{|N|} \tag{1}$$

Evaluate:

$$(n) = \begin{cases} 1\ if\ classify(n) = cn \\ 0\ else \end{cases} \tag{2}$$

where:

$N$ is the testing dataset to be classified.

$|N|$ represents the size of the testing data set to be classified

*Classify* ($n$) gives the classification result of the data item $n$ by the deep network.

Classification accuracy can be given also by the equation:

$$Accuracy = \frac{TP + TN}{TP + TN + FP + FN}\ (\%) \tag{3}$$

The test of the dataset was analyzed by the use of six classification algorithms, which were naïve Bayes, J48, ZeroR, random tree, OneR, and SMO. The accuracy level of all the algorithms is given below in Table 5.

**Table 5.** Glucose Data.

| Algorithms | Correctly Classified Instances | Incorrectly Classified Instances |
|---|---|---|
| Naïve Bayes | 85.1563% | 14.8438% |
| J48 | 99.1667% | 0.8333% |
| ZeroR | 68.9063% | 31.0938% |
| RandomTree | 98.75% | 1.25% |
| SMO | 80.6771% | 19.3229% |
| OneR | 99.1146% | 0.8854% |

From Table 4, which shows the comparison between algorithms like ZeroR, random tree, J48, naive Bayes, OneR and SMO (sequential minimum optimization) we see that the correctly classified instances are higher than the incorrectly classified instances in terms of accuracy. Figure 10 shows a bar graph that represents correctly and incorrectly ranked instances of the used algorithms.

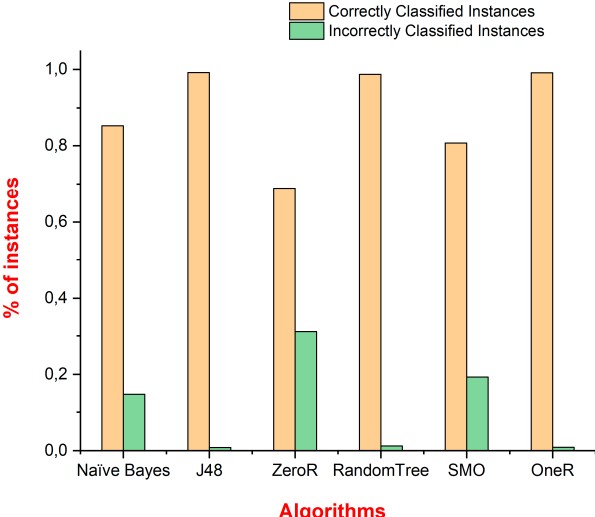

**Figure 10.** The graph of correctly and incorrectly classified instances of algorithms.

Figure 11 shows the time graph of various classification algorithms. The longest time is taken by ZeroR, at 0.05 s, and the shortest time is taken by random tree, at 0.01 s only.

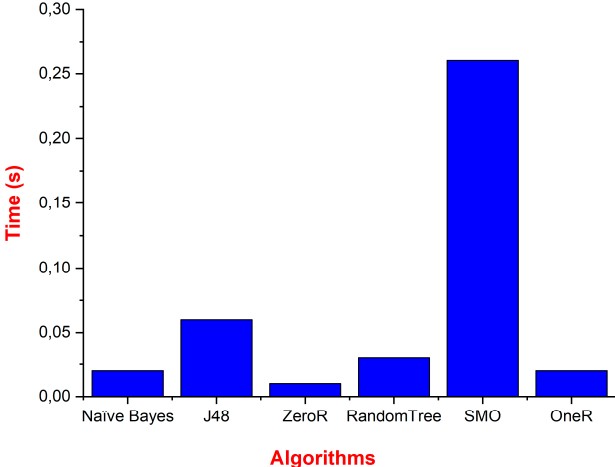

**Figure 11.** The graph for training time results for different algorithms.

As shown in Table 6, using parameters such as TP rate and PF rate, accuracy, recall, and F-measure can be calculated. These parameters are defined as follows:

$$\text{Precision} = TP/(TP + FP) \qquad (4)$$

$$\text{Recall} = TP/(TP + FN) \qquad (5)$$

$$\text{F-Measure} = (2 * \text{Precision} * \text{Recall})/(\text{Precision} + \text{Recall}) \qquad (6)$$

**Table 6.** Values of TP, FP, precision, recall, and F-measure for algorithms.

| Algorithms | TP Rate | FP Rate | Precision | Recall | F-Measure |
|---|---|---|---|---|---|
| Naïve Bayes | 0.852 | 0.319 | 0.871 | 0.852 | 0.837 |
| J48 | 0.992 | 0.012 | 0.992 | 0.992 | 0.992 |
| ZeroR | 0.689 | 0.689 | 0.475 | 0.689 | 0.562 |
| RandomTree | 0.988 | 0.019 | 0.987 | 0.988 | 0.987 |
| SMO | 0.807 | 0.419 | 0.839 | 0.807 | 0.778 |
| OneR | 0.991 | 0.013 | 0.991 | 0.991 | 0.991 |

Figure 12 shows the performance results of each classifier in terms of precision, recall, and F-measure.

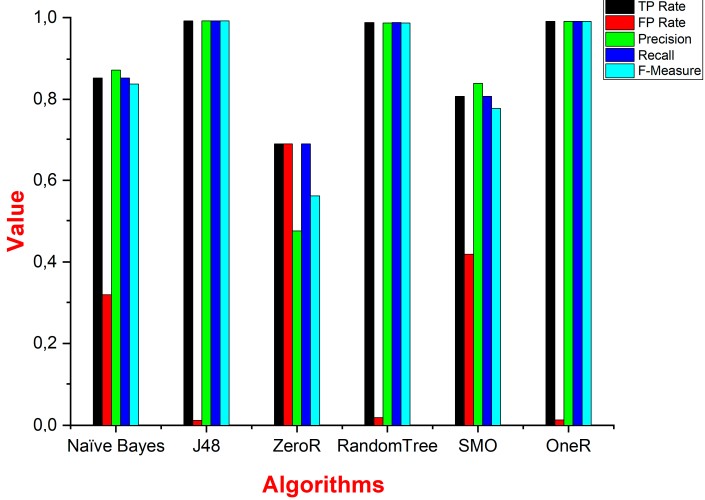

**Figure 12.** Graphical representation of FP, TP, precision, recall, and F-measure of different algorithms.

Using the preceding equations, we can also define specificity, sensitivity, precision, and accuracy. These parameters are given by the equations below:

$$\text{Specificity} = TN/(TN + FP) \qquad (7)$$

$$\text{Sensitivity} = TP/(TP + FN) \qquad (8)$$

$$\text{Precision} = TP/(TP + FP) \qquad (9)$$

$$\text{Accuracy} = (TP + TN)/(TP + TN + FN + FP) \qquad (10)$$

Table 7 shows the values of specificity, sensitivity, and accuracy for the algorithms used in this study.

**Table 7.** Comparison of sensitivity, precision, accuracy, and specificity.

| Algorithms | Specificity | Sensitivity | Accuracy | Precision |
|:---:|:---:|:---:|:---:|:---:|
| Naïve Bayes | 54.10% | 99.17% | 85.16% | 82.72% |
| J48 | 89.49% | 99.47% | 99.17% | 99.32% |
| ZeroR | 0.00% | 100% | 68.91% | 68.91% |
| RandomTree | 97.49% | 99.32% | 98.75% | 98.87% |
| SMO | 39.53% | 99.24% | 80.68% | 78.43% |
| OneR | 98.32% | 99.47% | 99.11% | 99.25% |

Figure 13 gives the graph of sensitivity, specificity, precision, and accuracy of different algorithms. The sensitivity, specificity, precision, and accuracy of OneR are better than those of the other classifiers.

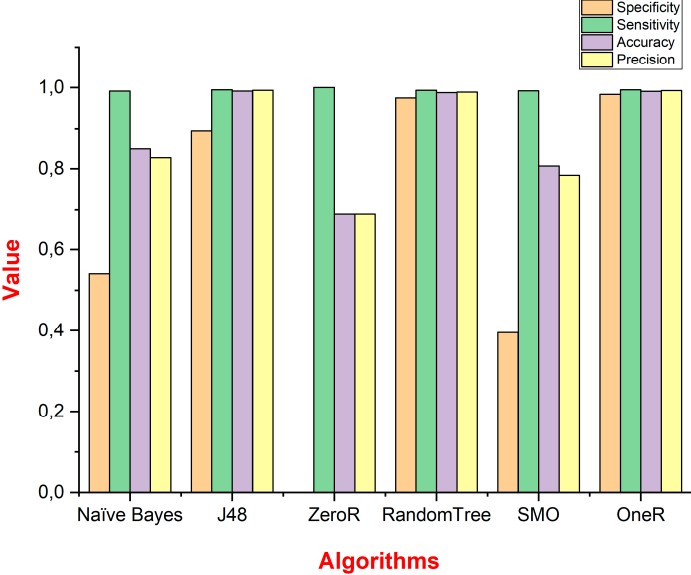

**Figure 13.** Graphical representation of sensitivity, specificity, precision, and accuracy of different algorithms.

Figure 14 represents the kappa statistic, MAE (Mean absolute error) and RMSE (root-mean-square error) values for the chosen classifiers. The ZeroR algorithm shows better results for MAE and RMSE, and the J48 algorithm shows a better value for the Kappa statistic.

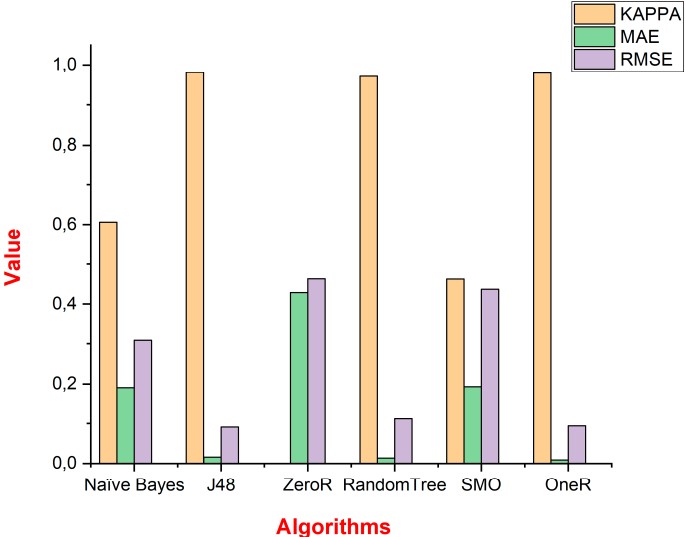

**Figure 14.** KAPPA Statistics and error rate values.

## 8. Conclusions

To conclude, the system which has been developed in this research can monitor the diabetic patient's physiological conditions, as proven by its high percentage accuracy in detecting normal glucose level for a diabetic patient. In this paper, an intelligent IoT system for remote monitoring of diabetic patients with a data classification model is presented for automated detection and classification of glucose level data. To improve the accuracy of analysis and diagnosis of diabetic patients, the system not only monitors health data such as blood sugar, but the proposed system offers advanced services such as interoperability, local storage, and data processing (classification of data using machine learning algorithms for prediction, and detection of falls via light algorithms). Additionally, after several tests, the proposed prediction system was evaluated by several machine-learning algorithms (naïve Bayes, J48, ZeroR, random tree, SMO, and OneR) and the simulation results demonstrated that the J48 algorithm exhibited excellent classification, with the highest accuracy of 99.17%, a sensitivity of 99.47% and a precision of 99.32%.

The system is also capable of producing a notification when an abnormal condition is detected. Our proposed model is expected to help diabetic patients and their family monitor their glucose level data from glucose level sensors using their smartphones. Additionally, the proposed model helps diabetic patients to obtain future predictions of their blood glucose levels. Therefore, with the proposed system, diabetic patients can avoid the worst conditions in the near future.

In the future, we will evaluate further parameters (including heart rate or any other factor that can be measured by wearables) for monitoring and will aim to implement the system in the modern smart healthcare field; we will also try to increase the number of patients for the test.

**Author Contributions:** Methodology and supervision, J.L. and A.O.; formal analysis and investigation, A.R., and M.H.; writing—original draft preparation, A.R.; writing—review and editing, A.R. and J.L. All authors have read and agreed to the published version of the manuscript.

**Funding:** This research received no external funding.

**Conflicts of Interest:** The authors declare no conflict of interest.

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
