# Peer review of "A Smart Glucose Monitoring System for Diabetic Patient"

_electronics, doi:10.3390/electronics9040678_

Round 1
Reviewer 1 Report
The focus of this paper is the design of a sensor device and the classification algorithm. As a research paper, the presented work lacks both technical depth and novelty. My concerns are summarized as follows:
- Technically, WiFi is not a good choice for your glucose with respect to energy efficiency. BT or BLE may provide you long battery life which can also satisfy the data communication requirements.
- Section 2 provides a review of the peer investigations. However, I find the authors just put the peer investigations there, and I am not clear why the presented work is necessary.
- I don’t find any novel work in section 5 communication protocol.
- What have the authors done regarding data classification?
- Equation (1) in section 7, the denominator should be N+1?
- The paper is not carefully written. The are many language errors:a.Line 26-27, Important IoTtechnical areas of IoT;b.Line 37, using these information’s improves;c.Line 40, several people?;d.Line 58, Smart devices such as smartphones and smart devices
Author Response
Dear Reviewer,
Thank you very much for your useful comments and for your time to help us to improve our paper.
Taking into account your comments. We have improved the introduction background. Moreover, in the conclusion section we have added the analysis of the results.
Comment 1:
The focus of this paper is the design of a sensor device and the classification algorithm. As a research paper, the presented work lacks both technical depth and novelty.
Reply 1:
In our paper, we describe the implementation of a new application of the Internet of Things in the healthcare field, which is used to use the new technologies of IoT and Big Data, to find a solution for diabetic patient. The aim of our paper is to discuss the implementation of a digital blood glucose measurement system integrated into the wireless network to monitor the glucose level in real-time. It is an intelligent IoT system for remote monitoring of healthcare. To improve the accuracy of the analysis and diagnosis of diabetic patients, the system not only monitors electronic health data such as blood sugar, but also patient movements, and also temperature. The monitoring system for diabetic patients based on a smartphone is developed. The blood sugar measurement with temperature and physical activity of the diabetic patient will be sent to the mobile phone via the wireless communication protocol (Wi-Fi, ESP8266). The selection of sensors is decided according to the vital parameter, which is responsible for monitoring the health of the diabetic patient.
The proposed system offers advanced services such as interoperability, local storage, and data processing (Classification of data using machine-learning algorithms for prediction, and anomaly detection using light algorithms).
These para graphs have been added in Section 2, page2.
Comment 2:
- Technically, WiFi is not a good choice for your glucose with respect to energy efficiency. BT or BLE may provide you long battery life which can also satisfy the data communication requirements.
Reply 2:
Energy efficiency is one of the crucial design factors. The medical sensor nodes must save their energy to operate for a few months or even a few years. Wi-Fi technology offers advantages over Bluetooth
We have also explained it in section 4 , page 6.
Comment 3:
- Section 2 provides a review of the peer investigations. However, I find the authors just put the peer investigations there, and I am not clear why the presented work is necessary.
Reply 3:
In section 2, we cite the new research that has developed IoT applications for patient monitoring, but No one of the revised systems are specially designed and developed for diabetic patient monitoring. There are many proposals for elderly people, and for disabled people. This work suggests a glucose level monitoring application for diabetic patients.
We have added a comparison table explaining the differences of our work with existing ones.
Comment 4:
- I don’t find any novel work in section 5 communication protocol.
Reply 4:
In section 5, we show the communication protocol of our application; we describe the essential steps of our application and especially the communication between the different elements of the application.
- The transmission between the sensors and the smartphone.
- The transmission from the smartphone to the database.
- The transmission from the database to the doctor.
Then we try to do a flow analysis of our application with the Wireshark during the transmission between ESP8266 and Arduino.
We have enhanced the contribution of this part in section 5, page 9.
Comment 5:
- What have the authors done regarding data classification?
Reply 5:
Regarding the classification of the data, we worked on the classification of the data of diabetic patients by several classification algorithms for the purpose of predicting the level of glucose in the blood for a diabetic patient based on the physical activity of the patient and also on the glucose level measured previously.
We have enhanced the contribution of this part in section 6, page 12.
Comment 7:
- Equation (1) in section 7, the denominator should be N+1?
Reply 7:
Please, notice that general the equation of the accuracy:
Accuracy = Number of correct prediction/Total of the cases to be predicted
So, Equation (1) in section 7, the denominator should be just N.
Comment 8:
- The paper is not carefully written. The are many language errors:a.Line 26-27, Important IoTtechnical areas of IoT;b.Line 37, using these information’s improves;c.Line 40, several people?;d.Line 58, Smart devices such as smartphones and smart devices
Reply 8:
We have improved language along the text
Reviewer 2 Report
The paper has an overall good lenght of both theoretical and practical content together with mathematical formulas that support the proposed algorithms.
What is missing is a comparison of the proposed system with other off-the-shelf IoT systems that are used for Glucose Monitoring systems.
Author Response
Dear Reviewer,
Thank you very much for your useful comments and for your time to review our paper.
Comment 1:
The paper has an overall good lenght of both theoretical and practical content together with mathematical formulas that support the proposed algorithms.
Reply 1:
Thanks for your comments
Comment 2:
What is missing is a comparison of the proposed system with other off-the-shelf IoT systems that are used for Glucose Monitoring systems.
Reply 2:
In section 2, we cite the new research that has developed IoT applications for patient monitoring, but no one of the mentioned research is specially designed and developed for diabetic patient monitoring. Several applications use Bluetooth but our application uses Wi-Fi for communication. There are many proposals for elderly people, and for disabled people. We do not find any applications that have already used or ready to use in the market for the glucose monitoring systems.
We have enhanced the contribution of this part in section 2, page 4. Moreover, we have added a comparison table to show the differences of our proposal with existing proposals.
Round 2
Reviewer 1 Report
The paper has been much improved according to the comments. My further questions are:
- Remove Figure 4. This is not a technical report.
- In Equation 2, From 0 to N, there are N+1 variables Evaluate(0) to Evaluate(N) !!!
Author Response
Dear reviewer,
Thank you very much for your time and for your detailed comments. Please, find below our reply.
Comment 1:
- Remove Figure 4. This is not a technical report.
Reply 1:
Figure 4 has been removed
Comment 2:
- In Equation 2, From 0 to N, there are N+1 variables Evaluate(0) to Evaluate(N) !!!
Reply 2:
In order to avoid any misunderstanding, in equation 1 we have corrected from i=1 to |N|.